# Resonant generation of propagating second-harmonic spin waves in nano-waveguides

K. O. Nikolaev [1,4], S. R. Lake[2,4], G. Schmidt [2,3], S. O. Demokritov [1] ✉ & V. E. Demidov [1]

Generation of second-harmonic waves is one of the universal nonlinear phenomena that have found numerous technical applications in many modern technologies, in particular, in photonics. This phenomenon also has great potential in the field of magnonics, which considers the use of spin waves in magnetic nanostructures to implement wave-based signal processing and computing. However, due to the strong frequency dependence of the phase velocity of spin waves, resonant phase-matched generation of second-harmonic spin waves has not yet been achieved in practice. Here, we show experimentally that such a process can be realized using a combination of different modes of nano-sized spin-wave waveguides based on low-damping magnetic insulators. We demonstrate that our approach enables efficient spatially-extended energy transfer between interacting waves, which can be controlled by the intensity of the initial wave and the static magnetic field.

Second-harmonic generation (SHG) plays an important role in modern technologies that use waves of different nature for transmission and processing of information. This phenomenon is particularly important in the field of photonics as it allows efficient generation of coherent optical waves that are difficult to generate directly. Because of its great practical importance, the optical SHG has been intensively studied over many decades[1–4], which has enabled the development of a wide variety of highly efficient photonic devices. Although SHG is a universal physical phenomenon, which can potentially be realized for any kind of waves, its practical realization requires fulfillment of two important conditions. First, SHG is possible only in media exhibiting nonzero second-order nonlinear susceptibility. Second, efficient SHG requires exact matching of the phase velocities (commonly referred to as phase matching) of the initial wave and the second-harmonic wave. In optics, the first requirement is met in a large class of non-centrosymmetric, nonlinear crystals. Thanks to the weak dispersion of light waves, the second condition of phase velocity matching can also be easily achieved by using a number of well-developed approaches[2–4].

In the case of magnetic systems, the requirement of nonzero second-order nonlinear dynamic susceptibility can be satisfied relatively easily. The second-order nonlinearity arises when the magnetization vector precesses in finite-size magnetic structures. Due to the dynamic demagnetization effects, the precession trajectory is typically elliptical under these conditions. This ellipticity initiates a dynamic magnetization component along to the precession axis at double the precession frequency[5]. In other words, magnetic SHG does not require the use of special media and can be observed in many magnetic materials and experimental configurations so long as there is elliptical magnetization precession[6–15]. This makes magnetic oscillations excellent candidate for generation of microwave-frequency harmonics. For example, in ref. 15 it was shown that the nonlinear response of magnetic domain walls to a driving magnetic field at megahertz frequencies can lead to the generation of up to 60 harmonics.

A significantly more complex task is the implementation of a phase-matched resonant SHG for propagating waves of dynamic magnetization (spin waves), which are believed to be one of the most promising candidates for nano-scale wave-based signal processing and computing[16–19]. In contrast to electromagnetic waves, spin waves exhibit strong dispersion, i.e., a strongly nonlinear dependence of the frequency on the wavevector. Accordingly, the phase velocity of spin waves changes significantly with the increase of their frequency, which

[1]Institute of Applied Physics, University of Muenster, 48149 Muenster, Germany. [2]Institut für Physik, Martin-Luther-Universität Halle-Wittenberg, 06120 Halle, Germany. [3]Interdisziplinäres Zentrum für Materialwissenschaften, Martin-Luther-Universität Halle-Wittenberg, 06120 Halle, Germany. [4]These authors contributed equally: K. O. Nikolaev, S. R. Lake. ✉e-mail: demokrit@uni-muenster.de

makes it difficult to achieve phase matching for waves at frequencies that differ by a factor of 2. As a result, in previous experiments with spin waves, only the non-resonant SHG process could be achieved, in which the generated second-harmonic wave did not belong to the eigenspectrum of spin waves and represented a forced non-resonant motion of magnetic moments, i.e., a non-propagating wave[6].

Recently it was shown theoretically[11] that phase-matched resonant SHG can be achieved in a semi-infinite magnetic film for spin waves propagating in a field-induced potential well near the edge of the film[20,21] (so-called edge modes) and bulk spin-wave modes of the film. However, to satisfy conservation of linear momentum, the second-harmonic waves must propagate away from the film edge, which reduces the efficiency of their interaction with the initial wave propagating along the edge. In addition, in real magnetic nano-systems, the edge modes exhibit enhanced spatial damping due to the scattering from edge imperfections (see, e.g., ref. 8), which strongly limits the practical applicability of the approach.

In this work, we propose and experimentally demonstrate a new approach that enables fully phase-matched, resonant generation of second-harmonic spin waves in magnetic nano-waveguides made from a low-damping magnetic insulator. We base this approach on the nonlinear interaction of spin-wave modes with different distributions of the dynamic magnetization through the thickness of the waveguide. We show that by choosing a proper thickness, one can engineer the dispersion spectrum of modes so that the phase velocities of a spin wave and its second harmonic become equal. Under these conditions, the initial spin wave continuously transfers its energy to the second-harmonic wave, resulting in a long-lasting, spatially-extended growth of the latter. This process can be achieved for different transverse modes and can be controlled by varying the intensity of the initial wave and the static magnetic field. Our experimental data show very good quantitative agreement with the results of theoretical analysis. The proposed approach provides new opportunities for the field of magnonics. It enables highly-efficient generation of spin waves with short wavelengths that are difficult to excite directly[22–27]. Due to the phase locking of the initial wave and its second harmonic, the approach can also be used for the implementation of new interference-based devices that utilize interference effects at the fundamental and doubled frequencies simultaneously.

## Results

### Studied system and approach

Figure 1a shows the schematics of the experiment. We study spin waves propagating in a waveguide with the width $w = 500$ nm fabricated from a film of Yttrium Iron Garnet (YIG)[28–33] with the thickness $d = 80$ nm. The spin waves are excited using a 500-nm wide and 200-nm thick Au antenna carrying microwave electric current. The waveguide is magnetized in plane by a static magnetic field $H$ applied perpendicular to its axis. The propagation of spin waves is analyzed with spatial and spectral resolution using micro-focus Brillouin light scattering (BLS) spectroscopy[34] (see Methods for details). This technique yields a signal, referred to as BLS intensity, which is proportional to the intensity of spin waves at the position, where the probing light is focused (Fig. 1a). This allows a direct imaging of spin waves with high spatial resolution. Thanks to the spectral resolution of the BLS technique, spin waves at different frequencies can be imaged independently. Additionally, we use the ability of BLS to detect the phase of propagating spin waves, which allows direct determination of their wavelength and phase velocity.

Figure 1b illustrates the main idea of our work – inter-mode resonant generation of second-harmonic spin waves. It shows the spectrum of spin-wave modes in a 500-nm wide YIG waveguide calculated using the analytical theory[35] and the approach developed in ref. 36 According to this approach, the frequency of spin-wave modes

can be calculated as:

$$f_{p,q}(k) = \frac{\gamma}{2\pi}\sqrt{\left[H + 4\pi M_s(1-P) + \frac{2A}{M_s}k_{tot}^2\right]\left[H + 4\pi M_s\frac{k^2}{k_{ip}^2}P + \frac{2A}{M_s}k_{tot}^2\right]}$$

(1)

Here, $k$ is the component of the wavevector along the axis of the waveguide, $\gamma$ is the gyromagnetic ratio, $M_s$ is the saturation magnetization, $P = \frac{k_{ip}^2}{k_{tot}^2} - \frac{k_{ip}^4}{k_{tot}^4}F\frac{1}{1+\delta_{0q}}$, $F = \frac{2}{k_{ip}d}\left(1-(-1)^q e^{-k_{ip}d}\right)$, and $A$ is the exchange constant. The effective wavevector in the plane of the waveguide is $k_{ip} = \sqrt{k^2 + k_z^2}$, where $k_z = \pi(p+1)/w$ is the effective wavevector characterizing the standing wave of the dynamic magnetization across the waveguide width[36] (see insets in Fig. 1b). The total effective wavevector is $k_{tot} = \sqrt{k_{ip}^2 + \kappa^2}$, where $\kappa = \pi q/d$ is the effective wavevector characterizing the standing wave of the dynamic magnetization through the thickness of the waveguide. This model assumes standard boundary conditions for dynamic magnetization[37]. The boundary conditions at the surfaces of the waveguide correspond to "unpinned" dynamic magnetization, which is typical for thin magnetic films. In contrast, the boundary conditions at the waveguide edges are determined by dipolar effects and correspond to "pinned" dynamic magnetization. The calculations are performed at $H = 500$ Oe using material parameters described in Methods.

The fundamental mode of the waveguide ($q = 0$, $p = 0$) is characterized by a uniform distribution of dynamic magnetization through its thickness (inset in Fig. 1b). This mode interacts most efficiently with the dynamic magnetic field of the antenna and, therefore, can be driven to a large-amplitude strongly-nonlinear regime using moderate powers of the excitation signal of the order of $10^{-4}$ W. As shown in the inset in Fig. 1a, in this regime, the ellipticity of the precession of magnetization **M** leads to the appearance of a sizable component of the dynamic magnetization along the z-axis. The projection of **M** onto the axis z, can be written as: $M_z = \sqrt{M_s^2 - m^2} \approx M_s\left(1 - \frac{1}{2}\frac{m^2}{M_s^2}\right)$, where $m$ is the dynamic component of the magnetization in the x-y plane. Substituting into this expression the dependence in the form $m^2 = m_x^2\cos^2(2\pi ft - kx) + m_y^2\sin^2(2\pi ft - kx)$, where $m_x$ and $m_y$ are the amplitudes of the dynamic magnetization in the $x$ and $y$ directions, respectively, we obtain:

$$M_z \approx M_s - \frac{1}{4M_s}\left(m_x^2 + m_y^2\right) - \frac{1}{4M_s}\left(m_x^2 - m_y^2\right)\cos(2\pi 2ft - 2kx) \quad (2)$$

The first two terms in Eq. (2) are time independent. The third term $m_z$ oscillates at frequency $2f$ while its spatial dependence is characterized by a wavevector $2k$. The amplitude of this magnetization component is proportional to $m_x^2 - m_y^2$, i.e. to the ellipticity of precession. Due to demagnetization effects in the narrow waveguide and the non-zero wavevector, this component creates a dynamic dipole magnetic field that is not strictly parallel to the static field $H$ and can linearly excite magnetization dynamics (see ref. 6). Similarly to $m_z$, this field oscillates with a frequency $2f$ and vary in space with a wavevector $2k$, where $f$ and $k$ are the frequency and the wavevector of the initially excited fundamental wave, respectively. Therefore, this dipole field is expected to couple the initial wave with the wave at $2f$ and $2k$.

This process can also be considered as the confluence of two magnons with energy $hf$ and linear momentum $hk$ into a magnon with energy $2hf$ and momentum $2hk$ according to the energy and the linear momentum conservation laws. We emphasize, however, that this process can be efficient only if the phase-space point ($2f$, $2k$) corresponds to an eigenexcitation of the system, which is difficult to implement in practice using one spin-wave mode. Three-magnon confluence processes can be easily realized when the initial magnons

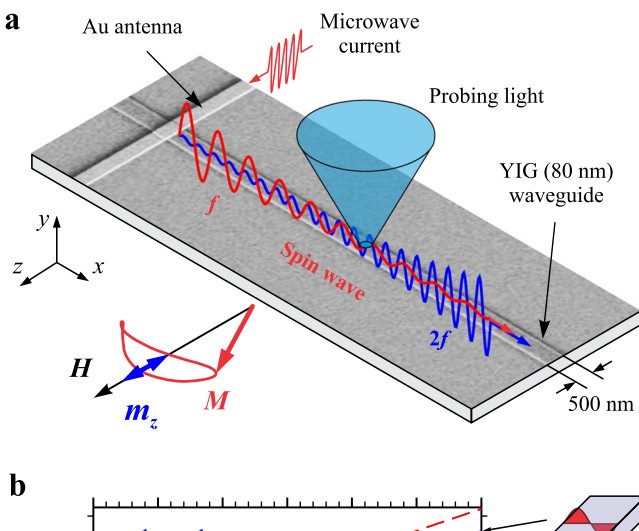

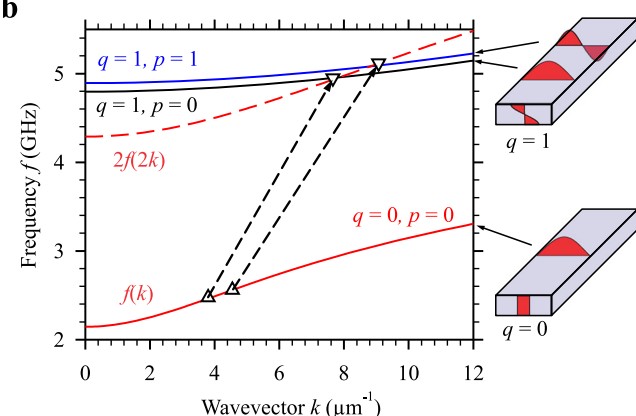

**Fig. 1 | Implementation of resonant generation of second-harmonic spin waves.**
**a** Schematics of the experiment. Spin waves in a 500-nm wide and 80-nm thick YIG waveguide are excited using a Au strip antenna. The ellipticity of the precession of the magnetization **M** leads to the appearance of a sizable double-frequency component of the dynamic magnetization $m_z \propto m_x^2 - m_y^2$ (inset), which results in the excitation of the second-harmonic spin wave. Both waves are independently detected by BLS. **b** Calculated dispersion spectrum of spin-wave modes. The phase-matching condition between the fundamental mode ($q = 0$, $p = 0$) and the first-order thickness modes ($q = 1$) is satisfied at the points corresponding to the intersection of the dashed curve $2f(2k)$ with the dispersion curves of the $q = 1$ modes. This enables a resonant energy transfer, as indicated by the dashed arrows. Symbols correspond to the resonantly interacting waves, as observed in the experiment. Insets schematically show the distributions of dynamic magnetization over the thickness and the width of the waveguide. The data are obtained at $H = 500$ Oe.

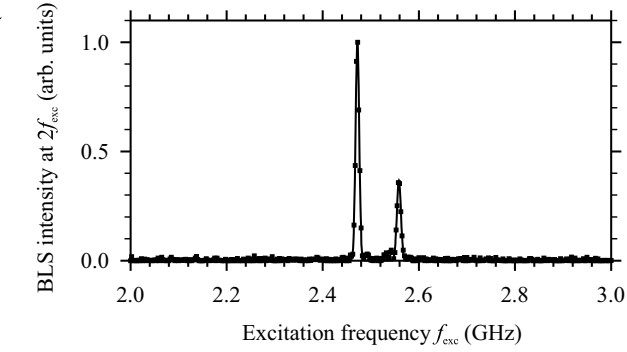

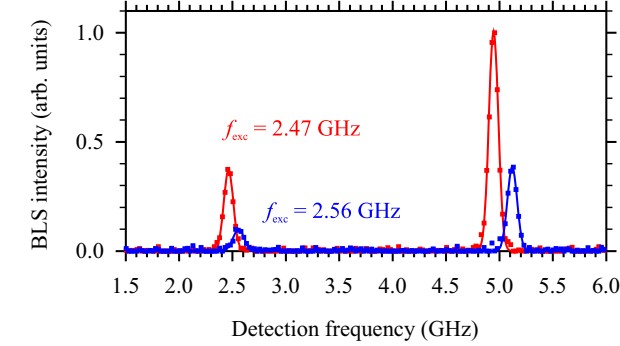

**Fig. 2 | Experimental evidence for resonant wave interaction. a** BLS intensity detected at a frequency twice the frequency of the initial spin wave $f_{exc}$ as a function of the latter. Note two narrow resonant peaks at $f_{exc} = 2.47$ and $2.56$ GHz.
**b** Complete BLS spectra recorded at two excitation frequencies corresponding to the observed resonances, as labeled. The data are obtained at $H = 500$ Oe at a distance $x = 10$ μm. Power of the excitation signal $P = 0.1$ mW.

have oppositely directed wavevectors, and the resulting magnon has a nearly zero wavevector[38,39]. In this case, the energy and momentum conservation rules can be easily satisfied even for waves, whose frequency vs wavevector dependence is not linear. However, to implement efficient resonant generation of the second harmonic, the three-magnon confluence process must involve two initial magnons with equal and collinear wavevectors and a resulting magnon with a doubled wavevector. Simultaneously, the frequency of the resulting magnon must be twice the frequency of the initial magnon. In contrast to the process involving counter-propagating magnons, such a process is difficult to implement due to the nonlinear dependence of the frequency of spin waves on the wavevector.

In order to find the frequencies at which $2f$ and $2k$ match magnon states in the sample, we plot the curve $2f(2k)$ on the dispersion diagram (red dashed line in Fig. 1b). Any intersection of the $2f(2k)$ curve with other magnon branches indicates a dedicated frequency for which SHG can become a highly efficient process. As seen from Fig. 1b, the dashed curve never intersects the curve for the fundamental mode ($q = 0$, $p = 0$). In other words, the required resonant condition cannot

be satisfied for a single spin-wave mode. However, as seen in Fig. 1b, in nanoscale YIG waveguides, the dashed curve can intersect with dispersion curves of first-order thickness modes ($q = 1$) characterized by a non-uniform distribution of dynamic magnetization through the thickness[40,41] (inset in Fig. 1b). Note that, by definition, the intersection points correspond to the condition of equality of the phase velocities $v_{ph} = 2\pi f/k$ of two spin-wave modes – the fundamental mode and the mode with doubled frequency. Therefore, it becomes possible to achieve a completely phase-matched inter-mode resonant energy exchange, as shown in Fig. 1b by arrows. We emphasize that for a given thickness of the waveguide, the resonant process is possible in a certain range of the static field $H$. By varying the field, one can shift the intersection points towards shorter or longer wavelengths.

## Evidence of resonant generation of second-harmonic spin waves
To prove the possibility of the resonant inter-mode process in practice, we first perform measurements at $H = 500$ Oe. We apply to the antenna an excitation signal at a frequency $f_{exc}$ varying from 2 to 3 GHz, which corresponds to the frequency range of the fundamental mode (Fig. 1b), and record the BLS signal at a frequency $2f_{exc}$ to observe possible SHG. Figure 2a shows the frequency dependence recorded at a distance $x = 10$ μm from the antenna. This curve exhibits two narrow peaks at $f_{exc} = 2.47$ and $2.56$ GHz, while the intensity found at other frequencies is below the noise background. This clearly shows that efficient second harmonic generation is possible only at specific frequencies, which indicates the resonant character of this process.

Figure 2b shows the complete BLS spectra recorded at two excitation frequencies corresponding to the observed resonances. These spectra allow one to simultaneously observe signals at the excitation frequency, as well as those corresponding to the second harmonic. The data show that, at $x = 10$ μm, the intensity of the second harmonic exceeds that of the initially excited wave by more than a factor of two

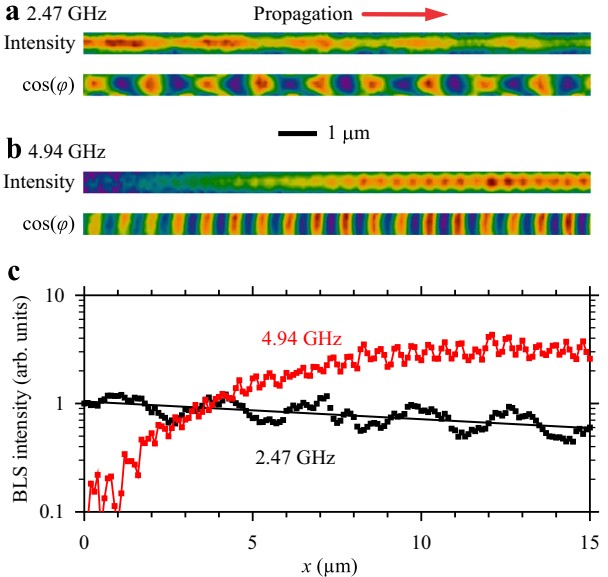

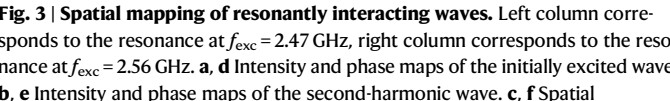

**Fig. 3 | Spatial mapping of resonantly interacting waves.** Left column corresponds to the resonance at $f_{exc} = 2.47$ GHz, right column corresponds to the resonance at $f_{exc} = 2.56$ GHz. **a, d** Intensity and phase maps of the initially excited wave. **b, e** Intensity and phase maps of the second-harmonic wave. **c, f** Spatial dependences of the intensity of the initial wave and the second-harmonic wave. Lines show the exponential fit of the data obtained for the initial wave. The data are obtained at $H = 500$ Oe. Power of the excitation signal $P = 0.1$ mW.

for both resonances. This indicates a very efficient energy transfer from the initially excited wave to the second-harmonic wave.

We study this processes in more detail in the space domain using the space- and phase-resolution of BLS. Figure 3 shows the results of spatial mapping of the intensity and phase ($\cos(\varphi)$) of spin waves corresponding to two observed resonances. Figure 3a–c characterizes the resonance at $f_{exc} = 2.47$ GHz, while Fig. 3d–f characterizes the resonance at $f_{exc} = 2.56$ GHz. Figure 3a shows spatial maps of the intensity and phase corresponding to the initial wave at 2.47 GHz, while Fig. 3b shows the same maps for the second-harmonic wave at 4.94 GHz. The intensity of the initial wave decreases with propagation distance, while the intensity of the second harmonic, which is negligible near the antenna, gradually increases in space. Figure 3c shows a direct comparison of the spatial dependences of the intensities of the initial wave and the second-harmonic wave. The initial wave exhibits a well-defined exponential decay (note the logarithmic scale of the vertical axis) characterized by the decay length of 54 μm (for comparison, the independently determined decay length of the wave at 4.94 GHz is 11 μm). The intensity of the second harmonic increases in the range $x = 0$–10 μm and then saturates. The intensities of the two waves quickly become equal at $x \approx 4$ μm.

Interestingly, at $x > 5$ μm, the intensity of the second harmonic becomes larger than the maximum intensity of the initial wave. We associate this with a difference in the group velocities of the two waves. In fact, although the waves have equal phase velocities, the second-harmonic wave possesses the group velocity of 0.2 μm ns$^{-1}$, which is 4 times smaller than the velocity of the initial wave of 0.8 μm ns$^{-1}$. The energy transferred by a wave is proportional to the product of intensity and group velocity. Therefore, when a wave with a large group velocity is converted into a wave with a smaller group velocity, the intensity must increase to satisfy the law of conservation of energy flux[42]. In agreement with this interpretation, the maximum intensity of the second harmonic does not exceed four times the maximum intensity of the initial wave. Note, however, that the ratio of these intensities is close to 4, reinforcing that there is high-efficiency energy transfer.

Analysis of the data obtained for the second resonance at $f_{exc} = 2.56$ GHz (Fig. 3d–f) demonstrates the main difference between the two observed resonances. Characteristics of the initial wave at 2.56 GHz (Fig. 3d) are not significantly different from those of the wave at 2.47 GHz (Fig. 3a). However, the spatial maps of their second harmonics differ substantially. While the wave at 4.94 GHz (Fig. 3b) is characterized by an intensity maximum in the center of the waveguide and a uniform distribution of phase across the waveguide width, the intensity of the wave at 5.12 GHz exhibits a minimum in the center and the phase shows a variation by π across the waveguide section. These differences indicate that the two resonances correspond to two different $q = 1$ modes that possess symmetric ($p = 0$) and antisymmetric ($p = 1$) transverse profiles.

This conclusion is in excellent agreement with the results of calculations (Fig. 1b). From the phase maps in Fig. 3, we obtain the wavelengths of the initial wave, $\lambda_O$, and the second harmonic, $\lambda_{SH}$, for the first ($\lambda_O = 1.65$ μm, $\lambda_{SH} = 0.82$ μm) and the second ($\lambda_O = 1.38$ μm, $\lambda_{SH} = 0.69$ μm) resonances. Taking into account the frequencies of these waves found from the previous analysis, we can plot the experimental points on the calculated dispersion diagram (symbols in Fig. 1b). As seen from these data, the point-up triangles, corresponding to the initial wave, coincide well with the dispersion curve of the fundamental mode, and the point-down triangles, corresponding to the second harmonic, are located at the intersections of the dashed curve with the dispersion curves of $q = 1$ modes with transverse quantization numbers $p = 0$ and 1, i.e., symmetric and antisymmetric transverse modes of the waveguide[34].

Comparison of the data of Fig. 3c, f allows us to draw one more important conclusion. The intensity of the second-harmonic wave at 5.12 GHz (Fig. 3f) grows with the propagation distance noticeably faster than that of the wave at 4.94 GHz (Fig. 3c). These data show that the second harmonic generation efficiency is higher for the mode $p = 1$. This is understandable, since the confluence of two magnons is also expected to cause a doubling of the transverse component of the wavevector, which favors generation of the mode $p = 1$. We also note, that the spatial decay of the initial wave at 2.56 GHz occurs faster than at 2.47 GHz. This is also the result of the faster energy transfer from the initial wave to the second harmonic due to the higher efficiency of the process. Generally speaking, the efficiency of the inter-mode SHG process is expected to be nonzero also for modes with $p > 1$. However, these modes possess a very short effective wavelength in the direction across the width of the waveguide and cannot be detected by our measurement setup.

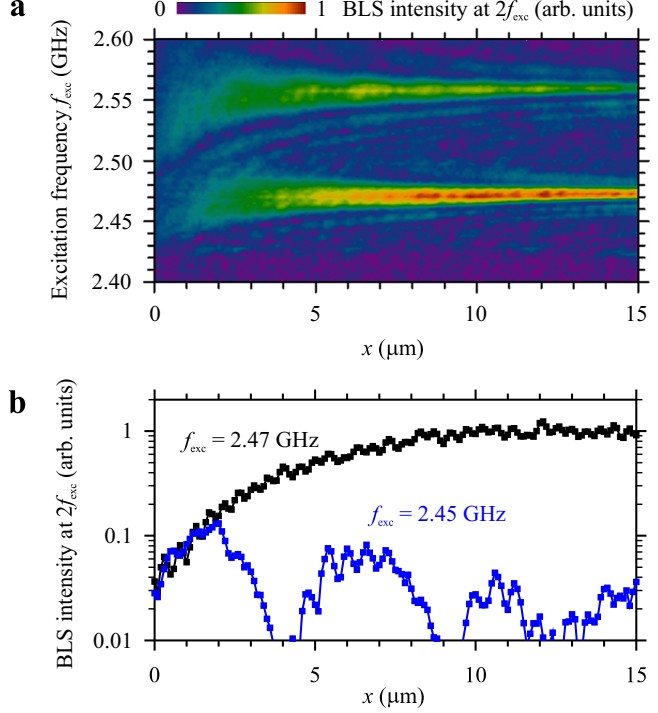

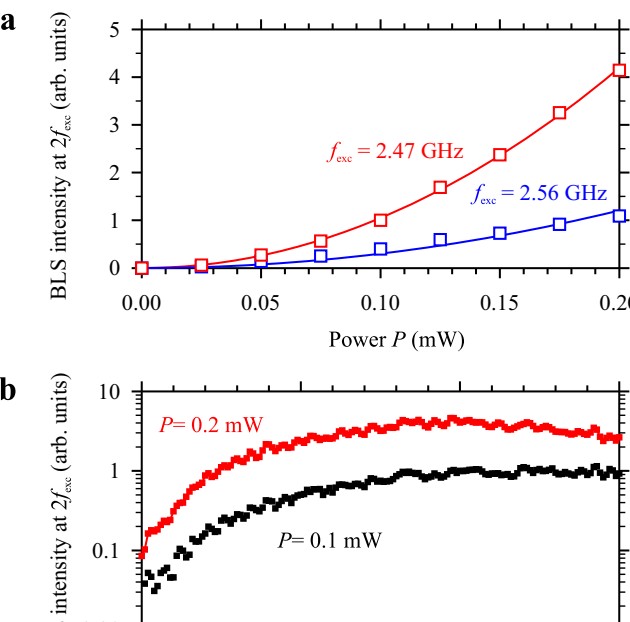

**Fig. 4 | Off-resonance interaction. a** Color-coded intensity of the second harmonic in frequency-space coordinates. **b** Spatial dependences of the intensity of the second harmonic recorded at $f_{exc}$ = 2.47 GHz (at resonance) and 2.45 GHz (out of resonance). The data are obtained at $H$ = 500 Oe. Power of the excitation signal $P$ = 0.1 mW.

**Fig. 5 | Dependence on the excitation power. a** Intensity of the second-harmonic wave for the two resonances, as labeled, as a function of the power of the excitation signal, $P$. The data are recorded at a distance $x$ = 10 μm. Symbols show experimental data. Curves show the fit by a parabolic function. **b** Spatial dependences of the intensity of the second harmonic recorded at $P$ = 0.1 mW and 0.2 mW, as labeled. The data are obtained at $H$ = 500 Oe.

## Off-resonance interaction

Let us now discuss the generation of the second harmonic at frequencies outside the resonances. We vary the excitation frequency $f_{exc}$ in the range 2.4–2.6 GHz around the found resonances and record spatial dependences of the intensity of the second harmonic at $2f_{exc}$. The obtained results (Fig. 4a) show that the spectral width of the resonant peaks strongly decreases with increasing propagation distance $x$. This is a natural feature of the resonant interaction, which requires phase matching between the initial wave and the second harmonic along the entire interaction path. At large interaction distances, the result becomes more sensitive to the difference in the phase velocities of the interacting waves. Correspondingly, efficient energy exchange can only be achieved over a narrow frequency interval. On the contrary, at small distances, a significant mismatch of phase velocities does not lead to a strong mismatch of the phases of the interacting waves, facilitating energy exchange over a wider spectral region. We emphasize, however, that the maximum achievable amplitudes of the second harmonic are much smaller in this case. This is demonstrated in Fig. 4b, which shows sections of the map Fig. 4a for the resonant frequency 2.47 GHz and an off-resonance frequency of 2.45 GHz. As seen from these data, in the region $x$ = 0–2 μm, the intensity of the second harmonic grows similarly for both frequencies. However, in the non-resonant case ($f_{exc}$ = 2.45 GHz), the intensity starts to decrease at $x$ > 2 μm until it completely vanishes at $x$ = 4 μm, only to return periodically for larger distances. This diminishment arises as the initial wave periodically becomes out of phase relative to the second-harmonic wave and, thus, suppresses it. The observed behaviors are similar to those found in optical systems[3], where the frequency dependence of the refractive index tends to lead to a phase mismatch between the initial wave and the second harmonic, unless special phase-matching approaches are used.

## Dependence on the excitation power

As discussed above, generation of the second harmonic relies on the component of the dynamic magnetization $m_z \propto m_x^2 - m_y^2$ (Fig. 1a), which, in the first approximation, is proportional the square of the amplitude of dynamic magnetization at the frequency of initially excited precession[5]. Therefore, we expect that the intensity of the second harmonic will be proportional to the square of the intensity of the initial wave. This is confirmed by the results presented in Fig. 5a, where we plot the intensity of the second-harmonic signal for the two resonances as a function of the power of the excitation signal, $P$, which is proportional to the intensity of the initial spin wave. As seen from Fig. 5a, the experimental data (symbols) are fit well with parabolic functions (solid curves). We note that the second harmonic generation is a non-threshold process that can be observed at arbitrarily small intensities of the initial wave. However, due to the nonlinear dependence of the intensity of the second harmonic on the intensity of the initial wave, it only becomes clearly pronounced at large intensities of the initial wave.

Figure 5b shows spatial dependences of the second harmonic intensity obtained for the first resonance ($f_{exc}$ = 2.45 GHz) at excitation powers $P$ = 0.1 and 0.2 mW. According to the aforementioned quadratic dependence, at a given distance $x \le 10$ μm, the intensity of the second harmonic increases by about four times when the intensity of the initial wave is doubled. This implies that the spatial rate of the energy transfer from the initial wave to the second harmonic increases with the increase in $P$. This greater efficiency gives rise to faster spatial attenuation of the initial wave at $P$ = 0.2 mW in comparison with $P$ = 0.1 mW. As a result, at this power, the energy transfer from the initial wave beyond $x$ = 10 μm can no longer fully compensate the attenuation of the second-harmonic wave and the intensity of the latter starts to decrease, in contrast with $P$ = 0.1 mW, where this decrease occurs at $x$ > 15 μm.

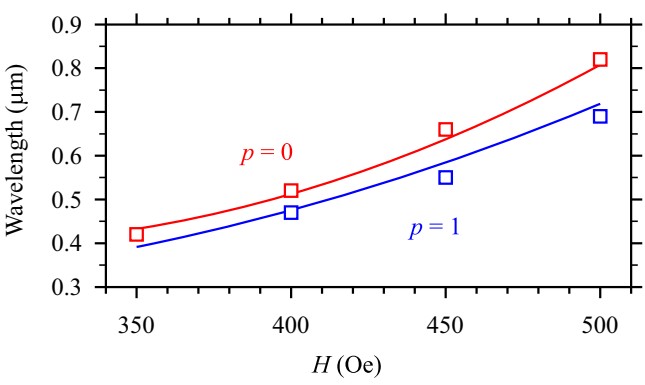

**Fig. 6 | Dependence on the static magnetic field.** Field dependences of the wavelengths of the second-harmonic waves for two resonances corresponding to $q = 1$ modes with transverse quantization numbers $p = 0$ and 1, as labeled. Symbols show experimental data. Curves show theoretical dependences obtained from calculations of dispersion spectra. Experimental data are obtained at $P = 0.1$ mW.

## Dependence on the static magnetic field

Finally, we discuss the effects of the static magnetic field $H$ on the studied phenomena. In the first approximation, a decrease in $H$ shifts the dispersion spectrum (Fig. 1b) down without significantly affecting the frequency gap between the fundamental mode and $q = 1$ modes, which is determined by the thickness of the magnetic film, as well as by its saturation magnetization and the exchange constant[35]. Under these conditions, the intersection points corresponding to the resonant interaction shift towards larger wavevectors (smaller wavelengths). To demonstrate this, we vary the static magnetic field, determine the resonant frequencies, and find the wavelengths of the second-harmonic waves from phase-resolved measurements. The results of these measurements are summarized in Fig. 6. The values of the wavelengths obtained from the experiment (symbols) are in good agreement with the results of analytical calculations (curves). As seen from Fig. 6, by decreasing $H$ to 350 Oe, second-harmonic waves with the wavelength below 500 nm can be resonantly generated. Such short waves cannot be excited directly by the used antenna[34]. In other words, the resonant second-harmonic generation process can be used to achieve efficient excitation of short-wavelength spin waves, which are difficult to excite using traditional inductive mechanism.

In conclusion, our results provide direct experimental evidence of highly efficient resonant second-harmonic generation by spin waves enabled by the engineering of the dispersion spectrum of spin waves in nanoscale YIG waveguides. This engineering allows one to fulfill the resonant conditions for the three-magnon interaction processes[38,39,43–45], in which the non-zero wavevector and the frequency of magnons can be doubled simultaneously. The demonstrated approach is flexible and can be customized for different microwave frequency ranges by simply varying the thickness of the magnetic waveguide. For example, for YIG waveguides with a thickness of 10–20 nm, the resonant conditions can be fulfilled for sub-THz band frequencies. The use of films with smaller thicknesses also makes it possible to tune the dispersion spectrum to achieve the fulfillment of the resonant conditions for the generation of higher-order harmonics. In addition to clear-cut and bountiful opportunities of generating high-frequency, ultra-short spin waves, resonant second-harmonic generation can also be used to implement novel magnonic devices. For instance, until now, magnonic devices exploiting wave-interference effects could only operate with signals carried by waves of the same frequency. The phase-locked second-harmonic generation process allows such devices to operate simultaneously at the fundamental and double frequencies. These possibilities will help to extend the functionalities of magnonic circuits and will propel new developments

within the field. Additionally, the demonstrated approach is fundamental and is not limited to spin waves. Indeed, engineering the inter-mode second-harmonic generation in other thin-film nanostructure systems may just grant the ability to exploit other types of waves (e.g. elastic waves) as well.

## Methods

### Sample fabrication

To fabricate the YIG waveguides, first a double-layer of PMMA resist was spin coated onto GGG $< 111 >$, then 8 nm of gold was evaporated to provide a conductive layer, and lastly the structures were patterned using e-beam lithography. Afterwards, the sample was placed in a potassium iodide solution to etch away the gold layer and then was developed in pure isopropanol. It was further processed with oxygen plasma to remove any remaining resist in the developed areas. Using the recipe established by Hauser et al.[32], nominally 100 nm of YIG was deposited at room temperature by pulsed laser deposition and lifted-off in acetone. The sample was annealed in oxygen for 3 h at 800 degrees, followed by a phosphoric acid etch to remove about 20 nm of YIG for precise thickness engineering and smoother edges. In order to do high frequency measurements on the sample, microstrip antennas had to be overlayed on top of the YIG waveguides. The same fabrication process used for the YIG waveguides, up until the PLD step, was used to pattern gold antennas on top. At this stage, 10 nm of titanium and 200 nm of gold were deposited by e-beam evaporation and then lifted off in acetone to complete the fabrication of the gold antennas.

### Micro-focus BLS measurements

Measurements are performed at room temperature. For the magneto-optical detection of propagating spin waves, we focus the probing laser light into a diffraction-limited spot on the surface of the YIG waveguide using a high-performance 100× microscope objective lens with a numerical aperture of 0.9. The probing light with a wavelength of 437 nm and a power of 0.25 mW is produced by a single-frequency laser. The spectrum of the light inelastically scattered from magnetic oscillations is analyzed using six-pass Fabry–Perot interferometer. The measured intensity of the scattered light is proportional to the intensity of spin waves. To obtain additional resolution with respect to the phase of spin waves, we use the interference of the scattered light with the light modulated by the signal used to excite spin waves. After processing, we obtain a value proportional to $\cos(\varphi)$, where $\varphi$ is the difference of the phase of the spin wave at the measurement position and the phase of the signal applied to the antenna.

### Calculation of the dispersion spectrum

We use the nominal geometrical parameters of the waveguide and standard for YIG saturation magnetization $4\pi M_s = 1750$ G. The exchange constant $A$ is used as an adjustable parameter. An excellent agreement between the experimental and the calculated dispersion is achieved in a broad range of the static magnetic field for exchange constant of 3.25 erg/cm, which is very close to the standard for YIG 3.66 erg/cm.

## Data availability

The data that support the findings of this study are available from the corresponding author upon reasonable request.

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

## Acknowledgements

This work was supported by the Deutsche Forschungsgemeinschaft (DFG, German Research Foundation) – project number 529812702 (G.S., V.E.D.). The work of K.O.N. was supported by the Deutsche Forschungsgemeinschaft (DFG, German Research Foundation) – Project-ID 433682494 – SFB 1459.

## Author contributions

K.O.N. performed measurements and data analysis. S.R.L grew and characterized the films, performed nanofabrication and data analysis. G.S., S.O.D. and V.E.D. formulated the experimental approach and supervised the project. All authors co-wrote the manuscript.

## Funding

## Competing interests

The authors declare no competing interests.
