## [Peer Review File · Nature Communications]

Reviewers' Comments:

Reviewer #1:

Remarks to the Author:

This is an interesting experimental manuscript where coherent generation of a second harmonic of a linearly excited magnonic signal was observed in a nano-scale (thickness -80 nm and width - 500 nm) rectangular ferrite waveguide.

The generation of the second harmonic of the external signal was realized in the process of a nonlinear interaction between the linearly excited magnonic modes and the discrete magnonic eigenmodes of a waveguide, quantized in frequency and wave vector due to the finite thickness and width of the waveguide.

The results are new and original, and the manuscript contains sufficient new physics to justify its publication in the "Nature Communications".

There are, however, several issues that are necessary to clarify before publication:

1. The interacting discrete magnonic modes of a waveguide have non-zero wavevector components along both the thickness and the width of the waveguide.

Judging from Fig.1, the authors assume the "unpinned" boundary conditions at the boundaries perpendicular to the thickness of the waveguide.

What are the boundary conditions for the dynamic magnetization at the boundaries perpendicular to the waveguide width?

2. It would be useful to present explicitly the conservation laws for the frequency and wavevector components of the discrete magnonic modes participating in the nonlinear process of the second harmonic generation. The authors claim that the second harmonic generation is caused by the second-order nonlinearity.

Usually, the second-order nonlinearity is a four-wave process with relatively small changes of frequency.

Most likely, the second harmonic generation reported here is a first-order three-wave confluence process.

3. The authors claim that the observed effect of the second harmonic generation can be used for the development of magnonic signal processing devices.

It would be useful to give a couple of examples of such nonlinear devices, and explain the advantages of such devices compared to the existing state-of-the-art.

In summary, the manuscript can be published in the "Nature Communications" after a minor revision.

Reviewer #2:

Remarks to the Author:

In the manuscript "Resonant generation of propagating second-harmonic spin waves in nanowaveguides", the authors reported the experiment of the generation of second harmonic spin waves in a magnetic waveguide with low damping. They achieved the phase matching condition between the spin wave and its second harmonic, and found that the initial spin wave continuously transferred its energy to the second harmonic wave with nonzero second-order nonlinear susceptibility. By using a phase resolved Brillouin light scattering microscopy, they directly imaged the spatial distribution of both initial and second harmonic spin waves and observed different spin-wave decay behaviors. The experimental results are consistent with the theoretical analysis considering a resonant confluence three-magnon process. The authors also applied different input

power and static magnetic fields to control the process of the generation of second harmonic spin waves. The proposed method might be useful for generating spin waves with short wavelength, as well as for new spin-wave interference devices.

The manuscript is well written, the experimental data taken by the BLS system are in very good quality, and the results are clearly presented. However, unfortunately I do not think the novelty of this work is sufficient for the publication in Nature Communications for the following reasons.

- A) The generation of higher order harmonic spin waves has been widely investigated in nonlinear spin-wave dynamics and even a series of spin-wave excitations up to the 60th harmonic of the excitation frequency has recently been observed [Science 375, 1165-1169 (2022)].
- B) Three-magnon confluence process can also be used to generate second order spin waves, which has been probed by wavevector selective Brillouin light scattering microscopy [Phys. Rev. B 79, 144428 (2009)] and recently been observed in a time-resolved manner in low damping YIG thin films [Phys. Rev. B 99, 024429 (2019)].
- C) Although the authors managed to fabricate a nanostructured magnetic waveguide based on a low damping YIG thin film which could provide a high generation efficiency, there is no significant difference in the physics of the second harmonic spin-wave generation compared to their previous work using permalloy [Phys. Rev. B 83, 054408 (2011)] and the work of others using CoFeB [J. Appl. Phys. 115, 053914 (2014)].

Therefore, I would recommend this work to be published in a more specific journal such as Communications Physics with only some minor suggestions as the following.

- 1) The equations of the spin-wave dispersion shown in Fig. 1b should be included in the manuscript.
- 2) In Fig. 3, the intensity of the second harmonic spin waves shows a growing trend and becomes much larger than the initial wave. I suggest that the authors extract the decay length of both initial and second harmonic spin waves for comparison.
- 3) Since the width of the nanowaveguide is in the sub-micrometer range, it is possible that the profiles are non-uniform and pinned at the waveguide edges due to dipolar interaction. I suggest that the authors add some discussion on this topic.

Reviewer #3:

Remarks to the Author:

The authors propose and experimentally demonstrate a mechanism for resonant excitation of second harmonic spin waves. Due to the magneto-dipolar interaction, the orthonormal spin wave modes with relatively large wavelength possess an elliptic trajectory which results in oscillations parallel to the magnetization direction at twice the frequency f of the mode. Interaction of this second harmonic oscillation of the directly excited spin wave with another magnon mode at frequency $2f$, can lead to resonant indirect excitation of the latter mode, if the matrix element of the interaction is nonzero, and the linear as well as angular momentum are conserved. To my understanding, the authors show that the dispersion of the nodeless modes across the waveguide width ($p=0$) and thickness ($q=0$) when multiplied by two and at k scaled by two, $2f_{\{p=0,q=0\}}(2k)$, intersects the dispersion of the modes with $q=1$ and $p=0,1$. The BLS spectroscopy experiments show that indeed the latter two modes are indirectly excited and spatially grow with distance from the stripline, while the directly excited spin waves attenuate, which signals a coherent energy transfer.

The authors also study the power dependence of the second harmonic signal, which is fitted well by a parabolic function supporting the phenomenology of the mechanism based on ellipticity of the directly excited spin waves. The authors also study the magnetic field dependence, and show the possibility of indirectly exciting small wavelength modes which can not be directly excited easily. The idea is interesting, the manuscript is generally well written and is quite accessible to a wide audience. A similar idea based on ellipticity of spin waves and a three field interaction for non-threshold excitation of resonances was used in Nat. Commun. 14, 490 (2023) to show sum and difference frequency excitation of NV center spin resonance by coherently mixing different spin

wave modes. Nevertheless, the authors achieve the second harmonic excitation within a monolithic magnet which is very useful. As listed below, I have some concerns about the underlying physics, some of the justification, and some of the choices in presentation, which I think should be addressed before I can recommend this work for a publication in Nature Communications.

(1) The proposed process behind the second harmonic generation expressed in magnon field operators is basically

$c_{(p=0,q=0),k}^{\dagger}c_{(p=0,q=0),k}^{\dagger}c_{(p=0(1),q=1),2k}+H.c.$, where c^{\dagger} is the annihilation (creation) operator. Such three magnon scattering is a result of the dipolar interaction. The authors can make the concept much clearer by using this kind of simple formulations. Currently, the reader might be confused that a resonant interaction between two orthonormal modes is possible. $c_{(p=0,q=0),k}^{\dagger}c_{(p=0,q=0),k}^{\dagger}+H.c.$ is the term in m_z which is due to the ellipticity of the spin waves. The authors have only mentioned $m_z \propto (m_x^2 - m_y^2)$ and have presented no mathematical clarification on the interaction leading to the second harmonic generation.

(2) In the current presentation, which modes are excited is not very clear. Does n refer to the number of nodes across the thickness or width or just a label? I tried to express my understanding of the modes excited in this work using p and q . I suggest the authors present a proper labeling convention.

(3) Why only two modes can be indirectly excited? Why there's such a hard cutoff? To my understanding, the process should remain effective, but with less efficiency, for larger number of nodes across the width. The efficiency should reduce because the matrix element of the three magnon interaction which contains spatial integration of the wavefunctions of the nodeless mode and $q=1$ modes reduces with increasing p .

(4) The discussions on the spatial dependence of the spin wave amplitudes are not very clear. Why smaller group velocity leads to larger amplitude at longer distances? When two wavepackets are excited and both have the same damping rate, the one with smaller group velocity would have a more substantial spatial attenuation. Therefore, the argument in the second paragraph of page 7 should be further elaborated.

(5) In the second paragraph of page 10, the justification of the downward trend of indirectly excited mode at $x > 10\mu\text{m}$, for $P=0.2\text{mW}$ is not clear. For $P=0.1\text{mW}$ too, the intensity should start to decrease at some point which hasn't been measured or plotted.

Response to Reviewer #1

The Reviewer writes:

This is an interesting experimental manuscript where coherent generation of a second harmonic of a linearly excited magnonic signal was observed in a nano-scale (thickness - 80 nm and width – 500 nm) rectangular ferrite waveguide.

The generation of the second harmonic of the external signal was realized in the process of a nonlinear interaction between the linearly excited magnonic modes and the discrete magnonic eigenmodes of a waveguide, quantized in frequency and wave vector due to the finite thickness and width of the waveguide.

The results are new and original, and the manuscript contains sufficient new physics to justify its publication in the “Nature Communications”.

Reply:

We thank the Reviewer for the positive evaluation of our work, for the constructive comments aimed at the improvement of our manuscript, and the recommendation to publish it in Nature Communications after a minor revision.

The Reviewer writes:

There are, however, several issues that are necessary to clarify before publication:

1. The interacting discrete magnonic modes of a waveguide have non-zero wavevector components along both the thickness and the width of the waveguide.

Judging from Fig.1, the authors assume the “unpinned” boundary conditions at the boundaries perpendicular to the thickness of the waveguide.

What are the boundary conditions for the dynamic magnetization at the boundaries perpendicular to the waveguide width?

Reply:

We agree with the Reviewer that the boundary conditions should be discussed in the manuscript. In fact, the model used in our manuscript assumes standard boundary conditions for dynamic magnetization. The boundary conditions at the surfaces of the waveguide correspond to “unpinned” dynamic magnetization, which is typical for thin magnetic films. In contrast, the boundary conditions at the waveguide edges are determined by dipolar effects and correspond to “pinned” dynamic magnetization.

In the revised manuscript, we have shown the corresponding distributions of the dynamic magnetization in Fig. 1b and added a discussion of the used boundary conditions on page 5. We have also cited the paper Phys. Rev. B 66, 132402 (2002) (new Ref. 29), which discusses the boundary conditions in thin magnetic stripes.

The Reviewer writes:

2. It would be useful to present explicitly the conservation laws for the frequency and wavevector components of the discrete magnonic modes participating in the nonlinear process of the second harmonic generation. The authors claim that the second harmonic generation is caused by the second-order nonlinearity.

Usually, the second-order nonlinearity is a four-wave process with relatively small changes of frequency.

Most likely, the second harmonic generation reported here is a first-order three-wave confluence process.

Reply:

We fully agree with the Reviewer that the second harmonic generation is a three-magnon confluence process, which is often called “first-order” magnon process. As is common in nonlinear physics, the term “second-order nonlinearity” is used in our manuscript to describe the nonlinearity associated with the existence of second-order magnetic susceptibility, which leads to the appearance of a double-frequency response. This nonlinearity is not related to the four-magnon (four-wave) interaction process.

In the revised manuscript, we have extended the discussion on pages 6-7 to make clear that the studied process is the three-magnon confluence.

The Reviewer writes:

3. The authors claim that the observed effect of the second harmonic generation can be used for the development of magnonic signal processing devices.

It would be useful to give a couple of examples of such nonlinear devices, and explain the advantages of such devices compared to the existing state-of-the-art.

Reply:

Complying with the Reviewer’s request, we have mentioned on page 12 of the revised manuscript possible ways to utilize the observed effect to extend the functionalities of magnonic devices. For instance, until now, magnonic devices exploiting wave-interference effects could only operate with signals carried by waves of the same frequency. The phase-locked second-harmonic generation process allows such devices to operate simultaneously at the fundamental and double frequencies.

The Reviewer writes:

In summary, the manuscript can be published in the “Nature Communications” after a minor revision.

Reply:

We hope that the Reviewer will find our revision satisfactory and will recommend publication of the revised manuscript in Nature Communications.

Response to Reviewer #2

The Reviewer writes:

In the manuscript "Resonant generation of propagating second-harmonic spin waves in nanowaveguides", the authors reported the experiment of the generation of second harmonic spin waves in a magnetic waveguide with low damping. They achieved the phase matching condition between the spin wave and its second harmonic, and found that the initial spin wave continuously transferred its energy to the second harmonic wave with nonzero second-order nonlinear susceptibility. By using a phase resolved Brillouin light scattering microscopy, they directly imaged the spatial distribution of both initial and second harmonic spin waves and observed different spin-wave decay behaviors. The experimental results are consistent with the theoretical analysis considering a resonant confluence three-magnon process. The authors also applied different input power and

static magnetic fields to control the process of the generation of second harmonic spin waves. The proposed method might be useful for generating spin waves with short wavelength, as well as for new spin-wave interference devices.

The manuscript is well written, the experimental data taken by the BLS system are in very good quality, and the results are clearly presented. However, unfortunately I do not think the novelty of this work is sufficient for the publication in Nature Communications for the following reasons.

Reply:

We thank the Reviewer for the positive evaluation of our work. We hope that the arguments given below will convince the Reviewer that our results represent a major step forward in the field and deserve publication in Nature Communications.

The Reviewer writes:

A) The generation of higher order harmonic spin waves has been widely investigated in nonlinear spin-wave dynamics and even a series of spin-wave excitations up to the 60th harmonic of the excitation frequency has recently been observed [Science 375, 1165-1169 (2022)].

Reply:

We emphasize that the process studied in the paper Science 375, 1165-1169 (2022) (Ref. 15) is completely different from that studied in our work. It is NOT the generation of higher-harmonic spin waves by a propagating spin wave. Instead, it represents the generation of frequency combs due to the strongly nonlinear response of domain walls to an external driving magnetic field at megahertz frequencies. To make this clear, we have added a comment on page 2 of the revised manuscript.

The Reviewer writes:

B) Three-magnon confluence process can also be used to generate second order spin waves, which has been probed by wavevector selective Brillouin light scattering microscopy [Phys. Rev. B 79, 144428 (2009)] and recently been observed in a time-resolved manner in low damping YIG thin films [Phys. Rev. B 99, 024429 (2019)].

Reply:

We fully agree with the Reviewer that the generation of second-harmonic spin waves can be described in terms of three-magnon confluence processes. This is discussed on page 6-7 of our manuscript. Such process can be easily realized when the initial magnons have oppositely directed wavevectors, and the resulting magnon has a nearly zero wavevector, which is the case considered in the papers mentioned by the Reviewer. In this case, the energy and momentum conservation rules can be easily satisfied even for waves, whose frequency vs wavevector dependence is not linear. However, to implement efficient generation of the second harmonic, as it is achieved in optics, the three-magnon confluence process must involve two initial magnons with equal and collinear wavevectors and a resulting magnon with a doubled wavevector. Simultaneously, the frequency of the resulting magnon must be twice the frequency of the initial magnon. Only in this case, one can achieve a spatially-extended synchronous energy transfer from the initial wave into the second-harmonic wave. In contrast to the process involving counter-propagating magnons, such a process has never been demonstrated for spin waves due to their nonlinear frequency vs wavevector dependence. In our

work, for the first time, we implemented this process using the peculiarities of the spin-wave spectrum in sub-100 nm thick YIG nano-waveguides.

To address the Reviewer's comment, we have extended the discussion of the magnon confluence process on pages 6-7 of the revised manuscript and cited the papers mentioned by the Reviewer (new Refs. 30 and 31).

The Reviewer writes:

C) Although the authors managed to fabricate a nanostructured magnetic waveguide based on a low damping YIG thin film which could provide a high generation efficiency, there is no significant difference in the physics of the second harmonic spin-wave generation compared to their previous work using permalloy [Phys. Rev. B 83, 054408 (2011)] and the work of others using CoFeB [J. Appl. Phys. 115, 053914 (2014)].

Reply:

We emphasize that there is a principal difference from the previous works mentioned by the Reviewer. In our work Phys. Rev. B 83, 054408 (2011), we reported the observation of *non-resonant* generation of the second harmonic. The generated wave of precession at double frequency did not belong to the eigenspectrum of spin waves and represented a forced non-resonant motion of magnetic moments, i.e., a non-propagating wave. Moreover, in previously studied Permalloy waveguides, it was impossible to implement a resonant process, since the spectrum of spin waves did not allow simultaneous fulfillment of the momentum and energy conservation conditions. The same is applicable for the work J. Appl. Phys. 115, 053914 (2014), which was mostly devoted to the electrical detection of the non-resonant process described in Phys. Rev. B 83, 054408 (2011). Note that the difference in efficiency between resonant and non-resonant processes is dramatic. This can be easily seen in Fig. 2a. The non-resonant process, which exists at all frequencies, results in negligible intensity of the second harmonic in comparison to the resonant process at the frequencies of the peaks.

In the revised manuscript, we have added a comment on page 3 to clarify these issues.

The Reviewer writes:

Therefore, I would recommend this work to be published in a more specific journal such as Communications Physics with only some minor suggestions as the following.

1) The equations of the spin-wave dispersion shown in Fig. 1b should be included in the manuscript.

Reply:

Following the Reviewer's recommendation, we have added on page 5 of the revised manuscript the equations used to calculate the spin-wave dispersion.

The Reviewer writes:

2) In Fig. 3, the intensity of the second harmonic spin waves shows a growing trend and becomes much larger than the initial wave. I suggest that the authors extract the decay length of both initial and second harmonic spin waves for comparison.

Reply:

As requested by the Reviewer, we have indicated the decay lengths of both waves on page 8 of the revised manuscript.

The Reviewer writes:

3) Since the width of the nanowaveguide is in the sub-micrometer range, it is possible that the profiles are non-uniform and pinned at the waveguide edges due to dipolar interaction. I suggest that the authors add some discussion on this topic.

Reply:

We completely agree with the Reviewer that the profiles of the dynamic magnetization across the width of the waveguide are non-uniform and should be pinned at the waveguide edges. Following the Reviewer's suggestion we have discussed the boundary conditions for the dynamic magnetization on page 5 of the revised manuscript and showed the spatial profiles in Fig. 1b.

Response to Reviewer #3

The Reviewer writes:

The authors propose and experimentally demonstrate a mechanism for resonant excitation of second harmonic spin waves. Due to the magneto-dipolar interaction, the orthonormal spin wave modes with relatively large wavelength possess an elliptic trajectory which results in oscillations parallel to the magnetization direction at twice the frequency f of the mode. Interaction of this second harmonic oscillation of the directly excited spin wave with another magnon mode at frequency $2f$, can lead to resonant indirect excitation of the latter mode, if the matrix element of the interaction is nonzero, and the linear as well as angular momentum are conserved. To my understanding, the authors show that the dispersion of the nodeless modes across the waveguide width ($p=0$) and thickness ($q=0$) when multiplied by two and at k scaled by two, $2f_{\{p=0,q=0\}}(2k)$, intersects the dispersion of the modes with $q=1$ and $p=0,1$. The BLS spectroscopy experiments show that indeed the latter two modes are indirectly excited and spatially grow with distance from the stripline, while the directly excited spin waves attenuate, which signals a coherent energy transfer.

The authors also study the power dependence of the second harmonic signal, which is fitted well by a parabolic function supporting the phenomenology of the mechanism based on ellipticity of the directly excited spin waves. The authors also study the magnetic field dependence, and show the possibility of indirectly exciting small wavelength modes which can not be directly excited easily. The idea is interesting, the manuscript is generally well written and is quite accessible to a wide audience. A similar idea based on ellipticity of spin waves and a three field interaction for non-threshold excitation of resonances was used in Nat. Commun. 14, 490 (2023) to show sum and difference frequency excitation of NV center spin resonance by coherently mixing different spin wave modes. Nevertheless, the authors achieve the second harmonic excitation within a monolithic magnet which is very useful. As listed below, I have some concerns about the underlying physics, some of the justification, and some of the choices in presentation, which I think should be addressed before I can recommend this work for a publication in Nature Communications.

Reply:

We thank the Reviewer for the positive evaluation of our work and for the constructive comments aimed at the improvement of our manuscript. Below, we respond in detail to all the Reviewer's questions and inquiries, and describe how they have been addressed in the revised

manuscript. We hope that the Reviewer will find our responses and revisions satisfactory and will recommend publication of the revised manuscript.

The Reviewer writes:

(1) The proposed process behind the second harmonic generation expressed in magnon field operators is basically

$c_{\{(p=0,q=0),k\}}^{\dagger}c_{\{(p=0,q=0),k\}}^{\dagger}c_{\{(p=0(1),q=1),2k\}}+H.c.$, where c (c^{\dagger}) is the annihilation (creation) operator. Such three magnon scattering is a result of the dipolar interaction. The authors can make the concept much clearer by using this kind of simple formulations. Currently, the reader might be confused that a resonant interaction between two orthonormal modes is possible.

$c_{\{(p=0,q=0),k\}}^{\dagger}c_{\{(p=0,q=0),k\}}^{\dagger}+H.c.$ is the term in $m_{\{z\}}$ which is due to the ellipticity of the spin waves. The authors have only mentioned $m_{\{z\}}\alpha(m_{\{x\}}^2-m_{\{y\}}^2)$ and have presented no mathematical clarification on the interaction leading to the second harmonic generation.

Reply:

In response to the Reviewer's comment, we have added a detailed mathematical description of the process leading to mode coupling on page 6 of the revised manuscript. In particular, we have clarified how the double-frequency magnetization component is formed and discussed the mechanism of coupling of eigenmodes via the dipolar field created by the double-frequency magnetization component. We have also extended the discussion on pages 6-7 describing second-harmonic generation in terms of the three-magnon confluence processes.

The Reviewer writes:

(2) In the current presentation, which modes are excited is not very clear. Does n refer to the number of nodes across the thickness or width or just a label? I tried to express my understanding of the modes excited in this work using p and q . I suggest the authors present a proper labeling convention.

Reply:

We agree with the Reviewer that the description of the excited modes was not entirely clear in the initial manuscript. In the revised manuscript, we use the labeling suggested by the Reviewer and show the corresponding mode profiles in Fig. 1b. We have also added the equations used to calculate the mode spectrum on page 5 of the revised manuscript.

The Reviewer writes:

(3) Why only two modes can be indirectly excited? Why there's such a hard cutoff? To my understanding, the process should remain effective, but with less efficiency, for larger number of nodes across the width. The efficiency should reduce because the matrix element of the three magnon interaction which contains spatial integration of the wavefunctions of the nodeless mode and $q=1$ modes reduces with increasing p .

Reply:

We agree with the Reviewer that modes with a larger number of nodes across the width should also be excited. In fact, the cutoff is associated with the general limitation of magneto-optical technique, whose sensitivity quickly decreases when half the wavelength of the spin wave becomes smaller than the diameter of the probing light spot. For our setup, the latter is

about 250-300 nm. This allows observation of modes with $p=0$ and 1, but does not allow one to detect modes with $p>1$.

Addressing the Reviewer's comment, we have clarified this on page 10 of the revised manuscript.

The Reviewer writes:

(4) The discussions on the spatial dependence of the spin wave amplitudes are not very clear. Why smaller group velocity leads to larger amplitude at longer distances? When two wavepackets are excited and both have the same damping rate, the one with smaller group velocity would have a more substantial spatial attenuation. Therefore, the argument in the second paragraph of page 7 should be further elaborated.

Reply:

We agree with the Reviewer that this issue needs more clarification. In fact, the SHG process is the conversion of the initial wave into a wave of doubled frequency. The energy transferred by a wave is proportional to the product of intensity and group velocity. Therefore, when a wave with a large group velocity is converted into a wave with a smaller group velocity, the intensity must increase to satisfy the law of conservation of energy flux. This was directly demonstrated for spin waves in Ref. 32.

Addressing the Reviewer's question, we have clarified this on page 9 of the revised manuscript.

The Reviewer writes:

(5) In the second paragraph of page 10, the justification of the downward trend of indirectly excited mode at $x>10\mu\text{m}$, for $P=0.2\text{mW}$ is not clear. For $P=0.1\text{mW}$ too, the intensity should start to decrease at some point which hasn't been measured or plotted.

Reply:

We thank the Reviewer for pointing out this issue. Now we see that our explanation was not very clear. Of course, at some point the intensity should start to decrease also at $P=0.1\text{ mW}$. This decrease is a natural result of finite damping. We would only like to state that this decrease occurs at smaller distances, with the increase in P .

We have modified the corresponding paragraph (page 12 in the revised manuscript) to make this clear.

Reviewers' Comments:

Reviewer #1:

Remarks to the Author:

In the opinion of this Referee, in the course of revision the authors addressed all the concerns of the Referees, and the revised version of their manuscript can be published in the Nature Communications as is.

Reviewer #2:

Remarks to the Author:

The authors have carefully revised the manuscript based on the referee reports and have very well addressed most of the referee questions, showing great expertise in the field and also solidity of their experimental findings. As the authors clarified, the resonant condition of the second harmonic spin-wave generation is of great importance and dramatically increases the efficiency compared to the forced non-resonant motion of magnetic moments. I'm now fully convinced by their arguments, and I appreciate their efforts to have significantly improved the quality of the paper. This work is highly specialised in the field of magnonics, and demands quite some knowledge of the field as well as rather careful reading of the results in order to appreciate its novelty and impact. The readability to the non-experts should still need some improvement. Before I can recommend its publication in Nature Communications, I would invite the authors to consider the following suggestions and to make some minor revision accordingly.

1) As the authors have stated, the effect studied in this work may be described in the context of three-magnon coupling, but is fundamentally different with conventional three-magnon coupling, where $(+k) + (-k) = 0$. [e.g. Phys. Rev. B 79, 144428 (2009); Nat. Mater. 10, 660 (2011); Phys. Rev. Lett. 125, 207203 (2020).; Phys. Rev. Lett. 130, 046701 (2023).] In this work, the authors achieved an anomalous type of three-magnon coupling with $k+k=2k$, which enables the nonlinear excitation of short-wavelength propagating spin waves. To me, this is the main novelty of this work, owing to which, this work deserves to be published in Nature Communications after some revision. Thus, I would suggest the authors to add one sentence, either in the introduction or in the conclusion to articulate this important point.

2) One of the key advantages of this studied effect, is to realise coherent excitation of propagating short-wavelength spin waves. The authors mentioned "short-wavelength spin waves that are difficult to excite directly,...". It would be helpful for the readers, if the authors could include some references on other methods exciting short-wavelength spin waves, e.g. directly [Science 322, 410-413 (2008); Appl. Phys. Lett. 109, 012403 (2016)] and indirectly [Nat. Commun. 9, 738 (2018); Nat. Nanotechnol. 14, 328-333 (2019).], etc.

3) It is critical to have the " $q=1$ " crosses with the $2f$ mode in the spin wave dispersion in order to trigger the resonant second-harmonic excitation. To "engineer" towards this aim, the high-quality YIG thin film is essential. I suggest the authors to stress this point while giving credits to pioneering works that make the high-quality nanometre-thick YIG films available for magnonics studies [e.g. IEEE Magn. Lett. 5, 6700104 (2014); Sci. Rep. 4, 6848 (2014)]. In addition, the " $q=1$ " mode may be considered as one kind of "perpendicular standing spin waves" that do not "stand", but instead exhibit a large in-plane wavevector and sizeable in-plane group velocity. This peculiar mode is very well described in previous studies [e.g. Phys. Rev. Lett. 122, 117202 (2019)]. It would be useful to the readers, if the authors can refer to some previous works when discussing the essential " $q=1$ " mode. The authors may want to mark in Fig. 1b the two high frequency modes as $(q=1, p=1)$ and $(q=1, p=0)$, and the low frequency one as $(q=0, p=0)$.

4) In optics, it is of great importance and interest to realise high harmonic generation (HHG), which on top of second harmonic generation (SHG), to generate third, fourth or even higher harmonics. The attosecond pulses of light, for instance, are generated through a mechanism based on the HHG plateau of femtosecond laser pulses. It would be very interesting for the broad readership of Nature Communications, if the authors could at least provide some perspectives for the extension of this studied technique towards HHG of magnons.

5) Some minor points: Page 5, when mentioning moderate power, the authors may provide an

approximate order of magnitude, otherwise readers from different fields may not have the same recognition of what power is "moderate". Page 6, "By varying the latter", may be more reader friendly to simply say "By varying the field", if I understand correctly.

Reviewer #3:

Remarks to the Author:

The authors have addressed all my concerns and comments to a satisfactory level. The manuscript is now significantly improved, and is more clear and rigorous to a degree that I can recommend its publication in Nature Communications as is.

Response to Reviewer #1

The Reviewer writes:

In the opinion of this Referee, in the course of revision the authors addressed all the concerns of the Referees, and the revised version of their manuscript can be published in the Nature Communications as is.

Reply:

We thank the Reviewer for the positive evaluation of our work and the recommendation to publish the revised manuscript in Nature Communications as is.

Response to Reviewer #2

The Reviewer writes:

The authors have carefully revised the manuscript based on the referee reports and have very well addressed most of the referee questions, showing great expertise in the field and also solidity of their experimental findings. As the authors clarified, the resonant condition of the second harmonic spin-wave generation is of great importance and dramatically increases the efficiency compared to the forced non-resonant motion of magnetic moments. I'm now fully convinced by their arguments, and I appreciate their efforts to have significantly improved the quality of the paper. This work is highly specialised in the field of magnonics, and demands quite some knowledge of the field as well as rather careful reading of the results in order to appreciate its novelty and impact. The readability to the non-experts should still need some improvement. Before I can recommend its publication in Nature Communications, I would invite the authors to consider the following suggestions and to make some minor revision accordingly.

Reply:

We thank the Reviewer for the positive evaluation of the revised manuscript and the recommendation to publish it after a minor revision. As described in detail below, we have taken into account all the Reviewer's requests.

The Reviewer writes:

1) As the authors have stated, the effect studied in this work may be described in the context of three-magnon coupling, but is fundamentally different with conventional three-magnon coupling, where $(+k) + (-k) = 0$. [e.g. Phys. Rev. B 79, 144428 (2009); Nat. Mater. 10, 660 (2011); Phys. Rev. Lett. 125, 207203 (2020).; Phys. Rev. Lett. 130, 046701 (2023).] In this work, the authors achieved an anomalous type of three-magnon coupling with $k+k=2k$, which enables the nonlinear excitation of short-wavelength propagating spin waves. To me, this is the main novelty of this work, owing to which, this work deserves to be published in Nature Communications after some revision. Thus, I would suggest the authors to add one sentence, either in the introduction or in the conclusion to articulate this important point.

Reply:

As suggested by the Reviewer, we have added a short discussion in the conclusion section articulating the mentioned point. We have also cited the papers mentioned by the Reviewer (Refs. 40,43-45).

The Reviewer writes:

2) One of the key advantages of this studied effect, is to realise coherent excitation of propagating short-wavelength spin waves. The authors mentioned “short-wavelength spin waves that are difficult to excite directly,...”. It would be helpful for the readers, if the authors could include some references on other methods exciting short-wavelength spin waves, e.g. directly [Science 322, 410-413 (2008); Appl. Phys. Lett. 109, 012403 (2016)] and indirectly [Nat. Commun. 9, 738 (2018); Nat. Nanotechnol. 14, 328-333 (2019).], etc.

Reply:

Following the Reviewer’s request, we have cited the mentioned papers along with other important works on the excitation of short-wavelength spin waves on page 4 of the revised manuscript (Refs. 22-27).

The Reviewer writes:

3) It is critical to have the “q=1” crosses with the 2f mode in the spin wave dispersion in order to trigger the resonant second-harmonic excitation. To “engineer” towards this aim, the high-quality YIG thin film is essential. I suggest the authors to stress this point while giving credits to pioneering works that make the high-quality nanometre-thick YIG films available for magnonics studies [e.g. IEEE Magn. Lett. 5, 6700104 (2014); Sci. Rep. 4, 6848 (2014)].

Reply:

Complying with the Reviewer’s suggestion, we have stressed on page 7 of the revised manuscript and in the conclusion section that the use of nanoscale YIG waveguides is essential to implement resonant excitation of the q=1 mode. We have also cited the mentioned papers along with other important works on the high-quality nanometre-thick YIG films on page 4 of the revised manuscript (Refs. 28-33).

The Reviewer writes:

In addition, the “q=1” mode may be considered as one kind of “perpendicular standing spin waves” that do not “stand”, but instead exhibit a large in-plane wavevector and sizeable in-plane group velocity. This peculiar mode is very well described in previous studies [e.g. Phys. Rev. Lett. 122, 117202 (2019)]. It would be useful to the readers, if the authors can refer to some previous works when discussing the essential “q=1” mode.

Reply:

As requested by the Reviewer, we have cited the mentioned paper (Ref. 39) along with the work, where the dispersion of q=1 was measured for the first time (Ref. 38), on page 7 of the revised manuscript.

The Reviewer writes:

The authors may want to mark in Fig. 1b the two high frequency modes as (q=1, p=1) and (q=1, p=0), and the low frequency one as (q=0, p=0).

Reply:

We have changed the labeling in Fig. 1, as recommended by the Reviewer.

The Reviewer writes:

4) In optics, it is of great importance and interest to realise high harmonic generation (HHG), which on top of second harmonic generation (SHG), to generate third, fourth or

even higher harmonics. The attosecond pulses of light, for instance, are generated through a mechanism based on the HHG plateau of femtosecond laser pulses. It would be very interesting for the broad readership of Nature Communications, if the authors could at least provide some perspectives for the extension of this studied technique towards HHG of magnons.

Reply:

We agree with the Reviewer, that our approach also enables generation of higher-order harmonics in films with smaller thicknesses. Following the Reviewer's suggestion, we have mentioned this in the conclusion section of the revised manuscript.

The Reviewer writes:

5) Some minor points: Page 5, when mentioning moderate power, the authors may provide an approximate order of magnitude, otherwise readers from different fields may not have the same recognition of what power is “moderate”. Page 6, “By varying the latter”, may be more reader friendly to simply say “By varying the field”, if I understand correctly.

Reply:

As recommended by the Reviewer, we have specified in the mentioned sentence (page 6 in the revised manuscript) the order of magnitude of the excitation power (10^{-4} W). We have also replaced “By varying the latter” with “By varying the field”.

Response to Reviewer #3

The Reviewer writes:

The authors have addressed all my concerns and comments to a satisfactory level. The manuscript is now significantly improved, and is more clear and rigorous to a degree that I can recommend its publication in Nature Communications as is.

Reply:

We thank the Reviewer for the positive evaluation of our work and the recommendation to publish the revised manuscript in Nature Communications as is.